# Spatial Generalization of Visual Imitation Learning with Position-Invariant Regularization

Zhao-Heng Yin[1], Yang Gao[2], Qifeng Chen[1]
HKUST[1]   Tsinghua University[2]
zhaohengyin@berkeley.edu

*Abstract*—How the visual imitation learning models can generalize to novel unseen visual observations is a highly challenging problem. Such a generalization ability is very crucial for their real-world applications. Since this generalization problem has many different aspects, we focus on one case called *spatial generalization*, which refers to generalization to unseen setup of object (entity) locations in a task, such as a novel setup of object locations in the robotic manipulation problem. In this case, previous works observe that the visual imitation learning models will overfit to the absolute information (e.g., coordinates) rather than the relational information between objects, which is more important for decision making. As a result, the models will perform poorly in novel object location setups. Nevertheless, so far, it remains unclear how we can solve this problem effectively. Our insight into this problem is to explicitly remove the absolute information from the features learned by imitation learning models so that the models can use robust, relational information to make decisions. To this end, we propose a novel, position-invariant regularizer called POINT for generalization. The proposed regularizer will penalize the imitation learning model when its features contain absolute, positional information of objects. Various experiments demonstrate the effectiveness of our method.

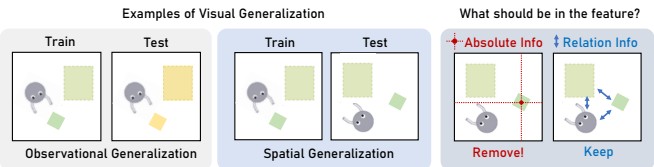

Fig. 1: **Left and Middle:** Two kinds of visual generalization. The examples are based on the MAGICAL benchmark provided by [15], in which a robot is required to relocate a box to a target region. The left figure shows an example of observational generalization, in which the only change during the testing phase is the visual texture of objects. The middle figure shows an example of spatial generalization. The object setup in the testing phase is unseen. **Right:** To achieve spatial generalization, we suggest that absolute information should be removed from the feature while the relational information should be kept. We propose a novel, position-invariant regularizer for this purpose.

## I. INTRODUCTION

Imitation learning is a class of algorithms that enable robots to acquire behaviors from human demonstrations [8]. The recent advance in deep learning has boosted the development of visual imitation learning and supported its applications like autonomous driving, robotic manipulation, and human-robot interaction [8].

In spite of its success, visual imitation learning methods still face many practical challenges. One major challenge is its ability to generalize to novel unseen visual observations, which is very common when we deploy the trained models [15, 11]. In the literature, this generalization problem is also known as the robustness problem. The problem covers many different aspects. For example, here we can identify two basic generalization capabilities: *observational generalization* and *spatial generalization* (Figure 1). Observational generalization refers to the generalization to novel visual textures. The changes in background color, object texture, or ambient light in the robotic manipulation task are examples of observational generalization. Such kind of visual change does not affect the underlying task structure (e.g., the position of object and targets) and only requires the robot to reason about semantic meanings correctly. In contrast, spatial generalization refers to the generalization to unseen setup of objects' (entities) locations in one task, which instead requires physical common sense about space and object. Consider the task of letting a warehouse robot move a box to some target region. If we set the initial position of the box to a place that is not covered by the demonstration dataset, then the imitation learning methods must be able to perform spatial generalization so as to succeed. In reality, the generalization challenge usually emerges as a combination of different generalization capabilities. In this paper, *we focus on the study of spatial generalization*.

For better spatial generalization, the visual imitation learning models should be able to obtain knowledge about objects and their spatial relations with proper inductive biases. Some work finds that vanilla deep visual imitation learning models strongly overfit to the absolute position of objects [15], which suggests that they do not extract relational information of objects to make decisions like humans [4]. Based on this observation, our main insight into this problem is to explicitly remove the absolute, positional information from the learned features in the visual imitation learning models. Note that this does not mean that the decision-making process is not dependent on absolute information. Rather, we expect that the model can extract the relational information (e.g., distance, direction) from the absolute information to make robust decisions. To this end, we propose a novel position-invariant regularizer called POINT. This regularizer will penalize the imitation learning model when it finds that the learned feature highly correlates with absolute, positional information. As a result, the imitation

learning model has to discover more robust relational features, and can generalize better in unseen scenarios.

## II. PRELIMINARIES

*a) Notations:* We model the sequential decision making problem as a Markov Decision Process $\mathcal{M} = (\mathcal{S}, \mathcal{A}, \mathcal{R}, \mathcal{T})$. $\mathcal{S}$ is the state space. $\mathcal{A}$ is the action space. $\mathcal{R}$ is the reward function. $\mathcal{T}$ is the transition dynamics. The agent's state at timestep $t$ is $s_t \in \mathcal{S}$. The agent takes action $a_t$ and receives reward $r_t = \mathcal{R}(s_t, a_t)$. Its state at timestep $t + 1$ is then $s_{t+1} \sim \mathcal{T}(s_t, a_t)$. The objective of the agent is to maximize the return $\sum_{t=0}^{T} \gamma^t r_t$, where $\gamma \in (0, 1]$ is a discount factor. For the imitation learning problem studied here, the agent has no access to $\mathcal{R}$ and $\mathcal{T}$, but it is provided with a fixed expert demonstration dataset $\mathcal{D} = \{\tau_i\}$. Here, each $\tau_i = (s_0^E, a_0^E, s_1^E, a_1^E, ... s_T^E, a_T^E)$ is an expert trajectory that can achieve high performance (return) in $\mathcal{M}$. Therefore, the agent should learn the behavior by leveraging the given demonstration dataset.

*b) Behavioral Cloning:* One classical imitation learning algorithm is the Behavioral Cloning (BC). BC turns the imitation learning problem into a supervised learning problem. It fits the expert's action $a_i$ given the observation $s_i$. For the visual imitation learning problem, the BC model can be divided into two consecutive parts: a vision encoder $f_\theta$ (which is usually a convolutional neural network), and a policy head $\pi$. The $f_\theta$ first encodes $s_i$ to the feature $f_i = f_\theta(s_i)$, and the $\pi$ then uses it to predict the expert's action. The BC algorithm minimizes the following negative log-likelihood objective:

$$\mathcal{L}_{BC} = \mathbb{E}_{(s_i, a_i) \in \mathcal{D}} \left[ -\log \pi(a_i | f_\theta(s_i)) \right]. \tag{1}$$

Due to its simplicity, BC is widely used in visual imitation learning. Therefore, we study the spatial generalization of BC in this paper.

## III. METHOD

### A. Formulation and Challenges

For the tasks that involve spatial generalization, there usually exist multiple objects in the observed states, such as the agent, the target object, and the goal. For the state $s_i$, we denote each of these objects in $s_i$ as $o_i^j$, and their positions as $(x_i^j, y_i^j)$. Then, our idea can be formulated as the minimization problem of each $I((\mathbf{x}^j, \mathbf{y}^j), \mathbf{f})$, where $I$ is the mutual information. Note that we use the notation $\mathbf{x}^j, \mathbf{y}^j, \mathbf{f}$ to indicate the corresponding random variables of $x_i^j, y_i^j, f_i$. However, this formulation leads to many practical challenges. First, since each $(x_i^j, y_i^j)$ is not provided directly by $s_i$ and should be inferred, we have to either train some object key-point detectors to detect the underlying objects in the training set, or annotate the objects by ourselves. However, both of these approaches can be difficult and tedious in practice. Second, even if we have ideal key-point detectors, we have to deal with a hard optimization problem in the summation form $\sum_j I((\mathbf{x}^j, \mathbf{y}^j), \mathbf{f})$. This can be intractable when there are many objects in the observed state.

Fortunately, we find that the previous works on the interpretation of deep learning models like GradCAM provide useful

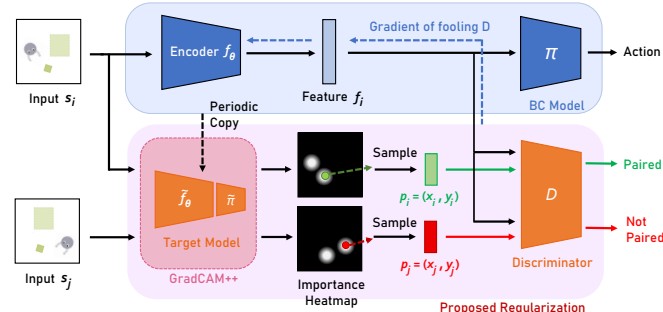

Fig. 2: Overview of our method. The blue branch above is the common imitation learning (BC) pipeline. Our proposed regularizer is shown in the light pink box at the bottom. The regularizer first uses the GradCAM++ algorithm to find out the important areas based on which the latest BC model makes decisions. Then it samples the coordinates from the discovered important areas and trains a discriminator network $D$ to calculate whether these sampled coordinates are paired with the feature $f_i$. The BC model (encoder $f_\theta$) is then trained to fool the discriminator $D$. When the encoder $f_\theta$ is able to fool $D$, the absolute positional information is removed from the feature as desired.

tools to handle these challenges. It can reduce the problem to a much simpler form. We discuss our observations as follows.

### B. Problem Reduction with GradCAM

GradCAM [13] is an interpretation method that can tell which part of the image is crucial in the decision process of a deep learning model. Given a BC model $(f_\theta, \pi)$ and input $s$, GradCAM outputs an importance heatmap of the same resolution as the input $s$. The heatmap indicates the importance of each pixel when we use this BC model for prediction. One nice property of this generated heatmap is that it is smooth and usually coincides with the meaningful objects in the input $s$. Therefore, we can consider the GradCAM as a rough object detector here.

We propose to sample $p_i = (x_i, y_i)$ from the generated heatmap, and then minimize the $I(\mathbf{p}, \mathbf{f})$. We find that this new objective can act as a proxy for the original objective in practice. Concretely, if $p_i$ is always far from a specific object like $o^k$, then we know that $o^k$ is irrelevant to the decision process of the current model. In this case, we conjecture that $I((\mathbf{x}^k, \mathbf{y}^k), \mathbf{f})$ should be low enough to meet our requirement. On the contrary, if $p_i$ always coincides with a certain object like $o^l$, then we actually minimize $I(\mathbf{p}, \mathbf{f}) \approx I((\mathbf{x}^l, \mathbf{y}^l), \mathbf{f})$ as we want.

### C. Loss Functions

Now, our remaining work is to reduce the mutual information $I(\mathbf{p}, \mathbf{f})$. However, we find that jointly estimating and minimizing the mutual information in our vision-based tasks is hard in practice. Since our ultimate goal is to minimize the information of $\mathbf{p}$ in $\mathbf{f}$, we instead propose an adversarial training framework to achieve this goal.

Specifically, we introduce a discriminator network $D$ to play a two-player min-max game with the BC model as follows.

$$\min_{f_\theta} \max_D \mathbb{E}_{(s_i,a_i)\sim\mathcal{D},(s_j,a_j)\sim\mathcal{D}} \quad (2)$$

$$[\log D(p_i, f_i) + \log(1 - D(p_j, f_i))]. \quad (3)$$

In this min-max game, the discriminator $D$ tries to tell the joint distribution of $\mathbf{p}$ and $\mathbf{f}$, denoted as $\mathbb{P}_{\mathbf{p},\mathbf{f}}$, from the product of their marginal distributions $\mathbb{P}_{\mathbf{p}\otimes\mathbf{f}}$. Meanwhile, the BC model is trying to fool the discriminator by removing the information of $\mathbf{p}$ from $\mathbf{f}$. Applying the convergence theory of the generative adversarial network (GAN) [6], we know that when $f_\theta$ is a global minimizer of Equation 2, $\mathbb{P}_{\mathbf{p},\mathbf{f}} = \mathbb{P}_{\mathbf{p}\otimes\mathbf{f}}$, which implies that $I(\mathbf{p},\mathbf{f}) = 0$. Therefore this min-max game fulfills our requirement.

In practice, we train $D$ to minimize the following binary classification loss function:

$$\mathcal{L}_D = -\mathbb{E}_{(s_i,a_i)\sim\mathcal{D},(s_j,a_j)\sim\mathcal{D}} \quad (4)$$

$$[\log D(p_i, f_i) + \log(1 - D(p_j, f_i))]. \quad (5)$$

However, for the encoder $f_\theta$, we find that using $-\mathcal{L}_D$ as the loss function for training will result in instabilities. We assume this is because the $f_i$ term is present in both of the two terms in Equation 2, which is different from that in the original GAN objective. Therefore, we propose to use the following loss function for optimization, which we find works well empirically:

$$\mathcal{L}_{reg} = \mathbb{E}_{(s_i,a_i)\sim\mathcal{D}}[\log D(p_i, f_i)]. \quad (6)$$

Combining the BC loss, the loss function to train the $f_\theta$ and $\pi$ is then

$$\mathcal{L} = \mathcal{L}_{BC} + \lambda \mathcal{L}_{reg} \quad (7)$$

$$= \mathbb{E}_{(s_i,a_i)\sim\mathcal{D}}[-\log \pi(a_i|f_\theta(s_i)) + \lambda \log D(p_i, f_i)]. \quad (8)$$

## IV. EXPERIMENTS

In the experiments, we first test the performance of our method on the MAGICAL benchmark. We study the generalization according to the IID protocol [9]. This means that the training and testing task distributions are the same, though the test instance will be unseen. Then, we provide an analysis of our algorithm through both qualitative and quantitive studies. Finally, we extend our method to a real robot manipulation problem.

### A. Task Setup

*a) MAGICAL:* The MAGICAL benchmark simulates a 2D robotic manipulation problem in a warehouse room. The tasks provided by the MAGICAL involve complex interactions between the agent and multiple objects, which require effective spatial generalization. In the experiments, we use a variant of its MatchRegion task. In this task, a robot is required to go across a square room to move some objects to a target region specified by a dashed rectangle. We set up several task instances of the MatchRegion task: MatchRegion-Target-1, MatchRegion-Target-2, MatchRegion-Target-2-Distract, MatchRegion-Target-3, MatchRegion-Target-3-Distract. We provide an illustration

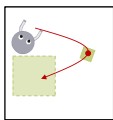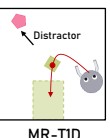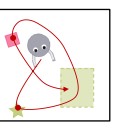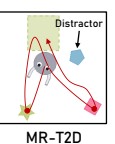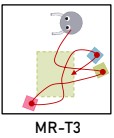

MR-T1    MR-T1D    MR-T2    MR-T2D    MR-T3

Fig. 3: The MAGICAL tasks used in our experiments. The grey robot is required to move the target objects (we mark them with red dots) to the target region. The red curve shows a possible plan to solve the task (the interaction details like releasing box are omitted). The long horizontal nature of this task brings additional challenges aside from the spatial generalization problem.

of these tasks in Figure 3. For each MatchRegion-Target-$X$ task (MR-T$X$), there is no distractor object in the room, so the robot only needs to move all the $X$ objects into the target location. However, for the MatchRegion-Target-$X$-Distract task (MR-T$X$D), there is an additional distractor object in the room. This object is also randomly placed in the room during testing. The existence of this distractor object not only increases the risk of learning spurious features but also adds to the difficulty of learning secure motions. As we will discuss later, even the existence of one distractor object can lead to a significant increase of generalization difficulty. The study of more distractors is carried out in the analysis part.

For each of the tasks above, we collect its human demonstration dataset by ourselves. For each demonstration trajectory, we randomly set up the initial position of the objects, target region, and the robot. For MR-T1, we collect 50 trajectories. For each of the other tasks, we collect 100 trajectories. The collection of all these trajectories takes 2 hours. We also study the outcome of using a different number of trajectories in the later analysis part.

### B. Baselines

For the vanilla BC policy, we train an IMPALA [5] policy, whose encoder is a residual convolutional neural network. We also try vision-transformer [3] and relational network [12] that have relational biases, but we find that they perform worse than IMPALA and do not report their results here. Then, we implement the following baselines for comparison: Dropout [14], Crop [17, 10], Cutout [2], MixReg [16], OREO [11], and CLOP [1].

### C. Results

*a) MAGICAL:* The result on MAGICAL is shown in Table I. The performance is defined by the success rate of the trained policy, which is the number of target objects that are successfully transferred to the target region, divided by the total number of target objects. We observe that our method is able to achieve state-of-the-art results and outperform the baselines by a large margin. Concretely, it improves the success rate by about 30%. Besides, we find that most of the previous regularization methods do increase the success rate of the vanilla version and their results are similar to each other. This shows that they

TABLE I: Evaluation result on the MAGICAL benchmark. We show the average score on three random seeds. Our method can achieve state-of-the-art results compared with the baselines.

| Method | Vanilla | Dropout | Crop | Cutout | MixReg | OREO | CLOP | Ours |
|---|---|---|---|---|---|---|---|---|
| MR-T1 | 0.09 ±0.02 | 0.28 ±0.04 | 0.42 ±0.03 | 0.19 ±0.03 | 0.26 ±0.02 | 0.21 ±0.03 | 0.16 ±0.06 | **0.63 ±0.05** |
| MR-T1D | 0.19 ±0.06 | 0.32 ±0.11 | 0.44 ±0.03 | 0.27 ±0.03 | 0.41 ±0.10 | 0.27 ±0.06 | 0.21 ±0.02 | **0.60 ±0.08** |
| MR-T2 | 0.25 ±0.03 | 0.48 ±0.03 | 0.46 ±0.04 | 0.43 ±0.05 | 0.44 ±0.05 | 0.37 ±0.05 | 0.32 ±0.07 | **0.75 ±0.07** |
| MR-T2D | 0.27 ±0.06 | 0.35 ±0.03 | 0.38 ±0.04 | 0.32 ±0.03 | 0.33 ±0.03 | 0.27 ±0.03 | 0.23 ±0.04 | **0.70 ±0.04** |
| MR-T3 | 0.23 ±0.02 | 0.51 ±0.03 | 0.47 ±0.05 | 0.32 ±0.04 | 0.48 ±0.05 | 0.42 ±0.04 | 0.35 ±0.07 | **0.66 ±0.03** |


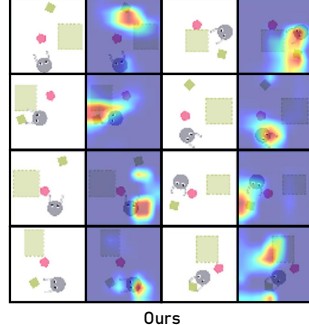

Dropout · Ours

Fig. 4: The GradCAM++ importance heatmap of the dropout model (left) and our model (right) on the MatchRegion-Target-1-Distract task. The red region indicates the most important region, while the dark blue indicates the least important region. The results suggest that the dropout model attends to the red distractor and is not robust. In contrast, our model is able to attend to correct objects.

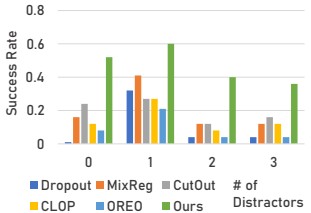

Fig. 5: The generalization performance to different number of distractors on MR-T1D.

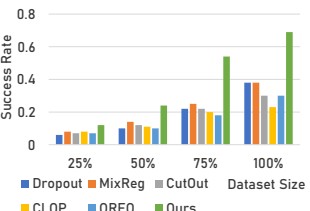

Fig. 6: The variation of performance on MAGICAL using the datasets of different sizes.

may solve some common issues in the generalization problem. However, their performance gap from our method suggests that we tackle a different issue here, which is overfitting to absolute positions.

### D. Analysis

*a) Qualitative Results:* To understand whether our method learns more robust features, we use GradCAM++ to visualize the learned model. For simplicity, we show the result on the MatchReigion-Target-1-Distract task. We compare the result of our model to the model trained with dropout here (Figure 4). We notice that the dropout model tends to focus on the red distractor object rather than the correct target object. In contrast, our model is able to focus on the correct objects. Even when the distance between the agent and the object is large, it can attend to the agent and the object simultaneously. The visualization results suggest that our regularizer indeed leads to robust relational features even when the vision network IMPALA does not have an explicit relational inductive bias. This accounts for the improvement of generalization.

*b) Unseen Number of Distractors:* A robust model should base its decision on robust relational information. As a result, for the MAGICAL tasks, it should be able to ignore the distractor and generalize to an unseen number of distractors. Therefore, we test whether our model trained on MR-T1D (where only one distractor presents) can generalize to MR-T1D with the unseen number of distractors (e.g., 0, 2, 3). We also compare the results with the previous models. The result is shown in Figure 5. We find that our model is able to generalize to the case of 0, 2, 3, though the performance is lower than the case of 1 (training scenario). In contrast, the prior model, such as the dropout model, fails in these unseen cases totally. This also echoes our qualitative analysis results.

*c) Number of Demonstrations:* We also study whether the proposed method works when the amount of expert demonstrations is limited. For this purpose, we test our method on the MAGICAL with $25\%, 50\%, 75\%$ of expert demonstrations. We show the averaged performance in Figure 6. We find that our method can achieve consistent improvement, though the performance decreases as the dataset becomes smaller. This result suggests that we still require a certain amount of diverse data to achieve spatial generalization.

## V. Conclusion

We studied the spatial generalization problem of imitation learning. We proposed POINT, a novel position-invariant regularizer to remove the absolute positional information from the features to tackle this problem. Through experiments on the MAGICAL benchmark as well as a robot manipulation system, we confirmed that previous methods do overfit to the absolute position and showed that our proposed approach can effectively help generalization.

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

## VI. APPENDIX

### A. Real-World Experiments

We also test whether our method scales to the real-world pick-and-place manipulation problem. We extend the MR-T1D to a UR10 robot arm with a Robotiq parallel-jaw gripper (Figure 7). As suggested by [7], we use a gripper camera and a workspace camera to provide observation. For the BC model, we use two separate IMPALA encoders to process each camera image, concatenate their output features along with the $z$-coordinate of gripper, and feed them into an MLP. We use the proposed regularizer to regularize the workspace branch. We collect 75 human demonstrations for training. We compare our method to dropout with different numbers of distract objects. The result is shown in Table II. Our method also achieves a large improvement in this problem. The qualitative results are shown in the Appendix Section VI-B.

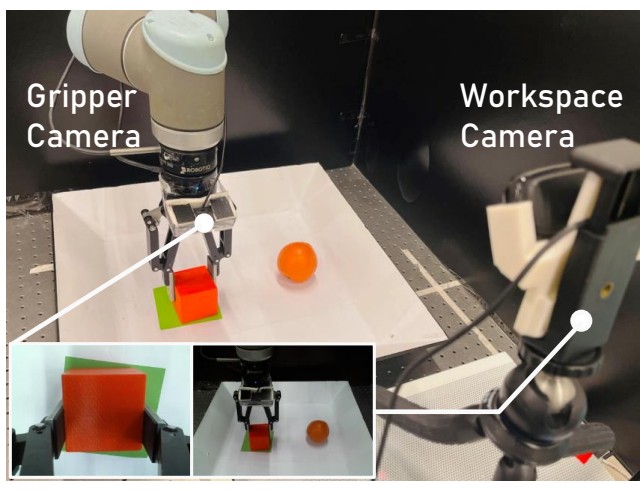

Fig. 7: The setup of real-world robot manipulation experiments.

TABLE II: The success rate of the real-world experiments. Our method is also effective here. Each test consists of 20 trials.

| Method | Dropout | Ours |
|---|---|---|
| 0 Dis. Obj | 35% | **55%** |
| 1 Dis. Obj | 35% | **60%** |
| 2 Dis. Obj | 20% | **50%** |
| 3 Dis. Obj | 10% | **45%** |

### B. Qualitative Results of the Manipulation Problem

In this section, we provide some qualitative results of the real-world manipulation problem. Recall that in this task, the robot is required to move a red cube to a target location specified by a green area. We show the importance heatmap of the dropout model (Figure 8) and our model (Figure 9). As is shown in the figures, we find that dropout model tends to attend more to the round distractor object compared with our model. However,

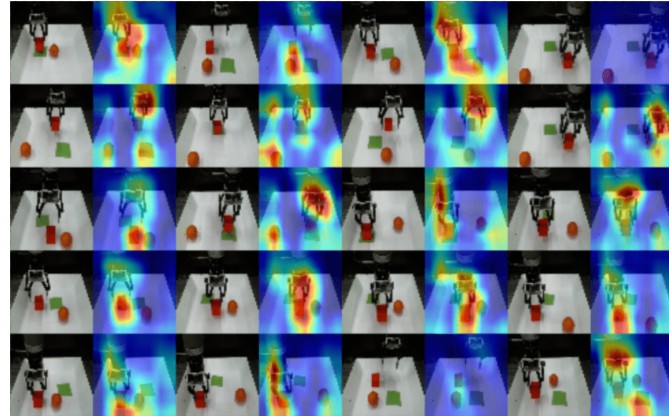

Fig. 8: The GradCAM++ importance heatmap of dropout model in the real-world manipulation problem. The dropout model tends to attend the round distractor object.

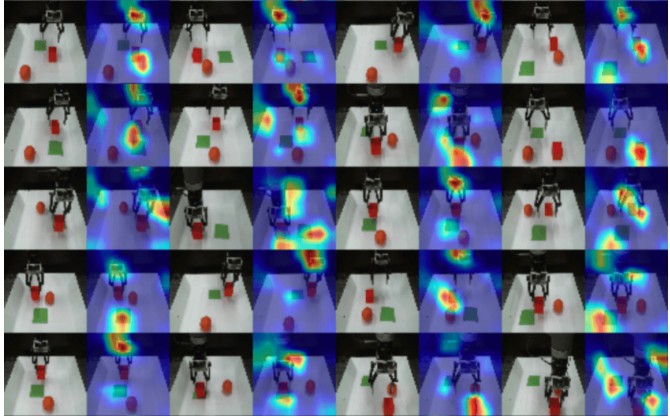

Fig. 9: The GradCAM++ importance heatmap of our model in the real-world manipulation problem. Our model attends less to the round distractor object. However, due to the visual complexity, we find that our model sometimes may attend the shadow in the background.

due to the visual complexity, we find that our model sometimes may attend the shadow in the background.