# OpenReview forum: "Spatial Generalization of Visual Imitation Learning with Position-Invariant Regularization"
_roboticsfoundation.org/RSS/2023/Workshop/Symmetry — RSS 2023 Workshop Symmetry_

### Official Review · Reviewer_eAe7 · 2023-06-10
**Strong Accept**

**Rating:** 9
**Confidence:** 5

**Review:**

This paper proposed a regularization method named POINT for behavior cloning to improve spatial generalization. The authors identify that  overfitting to absolute spatial information as a prominant anti-pattern for spatial generalization, which will be regularized using the proposed  method.
More specifically, POINT uses the mutual information of visual features and their corresponding pixel locations, weighted by GradCam pixel important heatmap, as the signal of overfitting to absolute position information. The authors then propose to jointly optimze the BC objective and overfitting signal as a minimax game, similar to GAN.
The authors demonstrated the advatange of using POINT over vannilla BC and other regularization methods on the MAGICAL 2D simulated manipulation benchmark.

The paper is clearly written with its core claimed backed up by extensive simulation experiments.

---

### Official Review · Reviewer_uRSc · 2023-06-12
**Review of Spatial Generalization of Visual Imitation Learning with Position-Invariant Regularization**

**Rating:** 7
**Confidence:** 4

**Review:**

## Summary
This paper proposes a new approach to learning features or representations from visual inputs such that the features disregard positional information while retaining relational information between objects or keypoints. The authors propose combining an encoder and an imitation learning policy with a discriminator that learns to generate coordinates of important regions in the image and pairs them with the features from the encoder. On a robotic manipulation benchmark with unseen object layouts, the proposed method performs better than other regularizing methods such as dropout, data augmentation, or other baselines.

## Strengths and weaknesses
Strengths: This paper focuses on the important problem of improving generalization, specifically generalization to new object layouts, from visual imitation learning. This is an important yet challenging problem and the authors propose a relatively simple yet effective approach to pair with an encoder + policy. Empirical results show a large improvement over other methods.

Weaknesses: A weakness of the paper is that it seems to only handle single-modal trajectories, i.e. there is a single position associated with each state/feature. This may restrict the policy actions to learn only a single optimal order of moving objects when there may be several. However, this is not a large restriction in practice and many existing imitation learning algorithms also exhibit this problem. Also, it would have been nice to verify that using GradCAM++ as a discriminator produces position-invariant features. Some baselines seem more like an ablation than a true comparison of different methods (i.e. dropout, cutout, etc.)

## Impact and novelty
The novelty of this paper is fair. It uses existing algorithms such as GradCAM++ or Impala but combines them in a novel way for visual imitation learning. It’s interesting that a vision transformer performs worse than GradCAM++, I would have expected an attention mechanism to do just as well or even better.

## Recommendation
I recommend this paper to be accepted as it proposes a simple yet effective approach for visual imitation learning and its contributions are relevant to the robotics community.

---

### Decision · Program_Chairs · 2023-06-23

**Decision:**

Accept

**Comment:**

Congratulations! We encourage the authors to revise the paper based on the reviewer's feedback.
Your paper will be presented as both a short presentation and a poster. Detailed instructions about the presentation format and camera-ready submission will be sent to you soon.